# Assessing COVID-19 Infection and Severe Disease Risk in Cancer Patients and Survivors: The Role of Vaccination Status, Circulating Variants, and Comorbidities—A Population-Based Study in Northern Italy

**DOI:** 10.3390/vaccines13121223

**Published:** 2025-12-03

**Authors:** Massimo Vicentini, Pamela Mancuso, Francesco Venturelli, Sergio Mezzadri, Eufemia Bisaccia, Alessandro Zerbini, Lucia Mangone, Paolo Giorgi Rossi

**Affiliations:** 1Epidemiology Unit, Azienda USL-IRCCS of Reggio Emilia, 42122 Reggio Emilia, Italy; massimo.vicentini@ausl.re.it (M.V.); pamela.mancuso@ausl.re.it (P.M.); francesco.venturelli@ausl.re.it (F.V.); lucia.mangone@ausl.re.it (L.M.); 2Infectious Disease Unit, Azienda USL-IRCCS di Reggio Emilia, 42122 Reggio Emilia, Italy; sergio.mezzadri@ausl.re.it; 3Public Health Unit, Azienda USL-IRCCS di Reggio Emilia, 42122 Reggio Emilia, Italy; eufemia.bisaccia@ausl.re.it; 4Unit of Clinical Immunology, Allergy and Advanced Biotechnologies, Azienda USL-IRCCS di Reggio Emilia, 42122 Reggio Emilia, Italy; alessandro.zerbini@ausl.re.it

**Keywords:** COVID-19, cancer, vaccination, hybrid immunity

## Abstract

**Background**: Cancer patients are particularly vulnerable to severe outcomes from COVID-19 due to immune suppression, treatment effects, and comorbidities. This population-based study aimed to assess how vaccination, circulating variants, and comorbidities influenced infections and severe disease risks in cancer patients compared with the general population. **Methods**: The study included 538,516 residents of Reggio Emilia Province, Italy, alive on 20 February 2020, followed until 30 September 2022. Cancer diagnoses (1996–2021) were obtained from the Reggio Emilia Cancer Registry and linked with COVID-19 surveillance, vaccination, hospitalization, and mortality data. Vaccination and prior infection were modelled as time-dependent variables. Hazard ratios for infection (HRs) and odds ratios for severe disease in those infected (ORs) were estimated using Cox and logistic regression models adjusting for sex, age, and comorbidities. **Results**: Among the 33,307 residents who had cancer, 9135 SARS-CoV-2 infections were recorded. Infection risk was similar to the general population before Omicron (HR 1.00; 95%CI 0.96–1.05) and slightly higher during Omicron (HR 1.08; 95%CI 1.05–1.11). Cancer patients showed higher probability of severe disease once infected (OR 1.33 pre-Omicron; 1.67 Omicron), with the greatest excess in recent diagnoses. Vaccination substantially reduced infections and severe outcomes in both groups in the pre-Omicron period; while only hybrid immunity reached high protection from Omicron infection. **Conclusions**: Vaccinations were effective in the populations with and without cancer; hybrid immunity conferred the strongest protection. However, because cancer patients, especially those recently diagnosed, retain a higher baseline risk of severe disease, vaccination yields even greater individual and public health benefits.

## 1. Introduction

The COVID-19 pandemic has posed significant challenges for global healthcare systems, placing a particularly heavy burden on vulnerable populations. Cancer patients, whether undergoing active treatment or in remission, represent a high-risk group for severe outcomes from COVID-19 due to a combination of factors, including immune suppression, cancer-related inflammation, reduced physiological reserve, organ damage, and treatment-related toxicities [1,2]. Emerging evidence has underscored the heightened risk of infection, hospitalization, and mortality in this population [3].

Cancer survivors face distinct risks associated with COVID-19. A population-based study from Reggio Emilia, northern Italy, conducted by our group [4] highlighted the higher cumulative mortality of COVID-19 among cancer survivors compared to the general population. Similarly, Johannesen et al. [5] conducted a population-based study in Norway, identifying specific risk factors that predispose cancer patients to severe outcomes from COVID-19. Their findings underscored the role of advanced age, comorbidities, and certain cancer types, such as lung cancer and hematologic malignancies, as significant predictors of adverse outcomes.

The risks of adverse COVID-19 outcomes in cancer patients are further modulated by vaccination status, the presence of comorbidities, and the emergence of circulating SARS-CoV-2 variants. Studies have demonstrated the efficacy of COVID-19 vaccination in reducing the severity of disease and mortality rates across various populations [6]. However, vaccine responses in cancer patients, particularly those undergoing immunosuppressive therapies, remain suboptimal, raising concerns about their level of protection against the virus [7].

Patients with cancer who received anticancer treatment within three months before a COVID-19 diagnosis experienced significantly worse outcomes compared to patients without cancer. These outcomes included a higher risk of mortality, intensive care unit (ICU) stay, and hospitalization. However, patients with cancer who had no recent treatment exhibited outcomes similar to or better than the general population, indicating heterogeneity within the cancer patient population [8].

Reinfections have emerged as a growing concern, especially with the appearance of immune-evasive variants like Omicron. We [9] observed that prior infection provided strong protection against reinfection in the pre-Omicron period, but this protection diminished significantly with the emergence of the Omicron BA.1 variant. Our findings showed that hybrid immunity, generated through vaccination and natural infection, offered substantially higher protection against reinfections compared to vaccination or prior infection alone. Furthermore, Omicron BA.1 reinfections were associated with reduced severity compared to primary infections, underscoring the evolving dynamics of reinfection risks in the context of new variants.

Emilia-Romagna, one of the earliest and most severely impacted regions during the pandemic, offers a unique population for assessing these risks. The region’s high COVID-19 burden and extensive vaccination campaigns offer valuable insights into the role of vaccines in mitigating risks among cancer patients and survivors. Furthermore, the emergence of SARS-CoV-2 variants with increased transmissibility and immune escape potential, such as the Delta and Omicron variants, has introduced additional complexities to understanding infection dynamics and vaccine efficacy in this subgroup [10,11].

Additionally, cancer patients often present with comorbidities such as cardiovascular diseases, diabetes, or chronic pulmonary conditions, which independently contribute to worse COVID-19 outcomes [12]. The interplay between cancer, comorbidities, and COVID-19 creates a multifaceted risk landscape that necessitates population-based studies to disentangle the relative contributions of these factors.

### Aim

This study aims to evaluate the risks of COVID-19 infection and severe disease in cancer patients in northern Italy, examining the role of vaccination status, circulating virus variants, comorbidities, and phase of cancer disease, compared with the general population.

## 2. Materials and Methods

### 2.1. Study Design and Population

This population-based cohort study included all residents of the province of Reggio Emilia, northern Italy, who were alive on 20 February 2020. The province has approximately 538,000 inhabitants. Through linkage with the Reggio Emilia Cancer Registry, individuals were classified according to whether or not they had a prior diagnosis of cancer between 1996 and 2021. The cohort was followed from 20 February 2020 to 30 September 2022 to identify SARS-CoV-2 infections, reinfections, and COVID-19-related outcomes. Demographic information, including residency status, age, and sex, was retrieved from the Population Registry of the Local Health Authority of Reggio Emilia. The study relied entirely on routinely collected data within institutional surveillance systems.

### 2.2. Data Sources

Data were obtained through record linkage of several administrative and clinical databases using the unique personal fiscal identification code. The Reggio Emilia Cancer Registry provided information on all incident malignant tumours diagnosed since 1996, including the date of diagnosis and tumour characteristics, classified according to international rules for multiple primary cancers. Following IARC/IACR international standards, the cancer registry records all incident malignant neoplasms (ICD-O-3 behaviour code/3), including both solid tumours and hematological malignancies. In accordance with international guidelines, non-malignant (benign and uncertain behaviour) tumours of the brain/central nervous system and of the urinary bladder are also included in incidence. For the present study, the cancer cohort included all individuals with a diagnosis of malignant cancer, excluding non-melanoma skin cancers, and non-malignant tumours of the brain/central nervous system and of the urinary bladder. In the presence of multiple primaries, the most recent tumour was selected for metachronous cases, whereas for synchronous tumours the most severe was retained. The registry operates with population-based completeness and standardized quality control procedures. Detailed methods used for the construction and validation of the cancer cohort are available in Mangone et al. [4].

The COVID-19 surveillance registry, coordinated by the Italian National Institute of Health, included all microbiologically confirmed SARS-CoV-2 infections detected in the province between February 2020 and September 2022. The same infrastructure was used to identify reinfections, defined as new positive tests occurring more than 90 days after a previous infection. The procedures for data collection, record linkage, and the definitions of infections and reinfections have been described in detail by Vicentini et al. [9].

The Vaccination Registry provided information on the date and type of each COVID-19 vaccine dose administered since the beginning of the national vaccination campaign in December 2020. The integration of these registries formed a comprehensive population-based surveillance system for infection, reinfection, and vaccination in Reggio Emilia, following the methodology described by Vicentini et al. [9].

Comorbidities, hospitalizations, and deaths were retrieved from hospital discharge records, the diabetes registry, and the mortality database of the Local Health Authority, and summarized using the Charlson comorbidity index (CCI). Cancer was excluded from the computation of the Charlson comorbidity index to avoid circularity when comparing individuals with and without a cancer diagnosis. During the study period, testing policies in Italy evolved from the exclusive use of RT-PCR assays to include third-generation antigen tests and, later, self-administered rapid tests that could be uploaded to an online platform for official confirmation. The definition of severe COVID-19 relied on hospitalization with SARS-CoV-2 infection; diagnosis and causes of hospitalization were not available in the COVID-19 information systems. The information about intensive care unit stay was not used because it was not uniformly collected during the period.

### 2.3. Exposure Definition

The exposure was a diagnosis of cancer, considered a time-dependent variable. Cancer patients were classified as exposed starting from the date of cancer diagnosis, while individuals without cancer contributed person-time to the unexposed group until a cancer diagnosis occurred, if ever. Cancer patients were further categorized according to time since diagnosis, which served as a proxy for disease phase and treatment status.

Those diagnosed less than two years earlier, who have the highest probability of being in the active therapy phase or in early follow-up.

Those diagnosed between two and five years earlier, who are patients probably still under oncologic surveillance but receiving no treatment or low-intensity treatments.

Patients diagnosed more than five years before, who are cancer survivors with the highest probability of being cured.

This time-dependent variable was used to evaluate how cancer status influenced infection and severity risks throughout the observation period.

A sensitivity analysis was conducted considering a 14-day lag time from vaccine dose administration and protection for the first and second dose, as previously proposed [13].

### 2.4. Outcomes

The primary outcome was a laboratory-confirmed SARS-CoV-2 infection recorded in the COVID-19 surveillance registry. Reinfections were defined as new positive tests occurring more than 90 days after the first infection, in accordance with national and European guidelines. Secondary outcome was severe COVID-19, given the infection, defined as hospitalization within 28 days or death within 90 days following diagnosis.

### 2.5. Follow-Up

For the first infection, the follow-up began on 20 February 2020 and ended at the first occurrence of infection, death, or the end of study period, i.e., 30 September 2022. For reinfections, follow-up started 90 days after the first positive test and continued until reinfection, death, or the end of the study period; the 90-day period after infection was considered as not at risk of infection. Hospitalizations were attributed to COVID-19 if they occurred from 3 days before up to 28 days after diagnosis, while deaths were considered COVID-19-related if they occurred within 90 days and met the official criteria for COVID-19 as the main cause of death. Therefore, COVID-19-related hospitalisations were included if they occurred up to 28 October 2022, while COVID-19-related deaths were included if they occurred up to 31 December 2022. For severe disease, date of disease onset was that of infection and not that of the hospitalization or death.

Analyses were divided into two periods, the pre-Omicron period (ending on 20 December 2021) and the Omicron period (beginning on 1 January 2022), excluding the transitional days between these dates to reduce misclassification; the proportion of variants in our province in the transition period are reported in Appendix A [14]. Covariates included sex, age, vaccination history, and comorbidities as measured by the Charlson comorbidity index. Age, vaccination, and prior infection were treated as time-dependent variables, updated daily during follow-up. All other covariates were assessed at baseline, corresponding to 1 January 2020.

### 2.6. Statistical Analysis

A descriptive analysis was performed to summarize the baseline characteristics of the study cohort, including sex, age, vaccination status, and Charlson comorbidity index (CCI), as well as the occurrence of SARS-CoV-2 infections over time expressed in person-days. For the cancer population, SARS-CoV-2 infections were also reported according to the number of years since diagnosis (<2, 2–5, and >5 years).

Cox proportional hazards models were used to estimate hazard ratios (HR) with 95% confidence intervals (95% CIs) for SARS-CoV-2 infection for the population with cancer, adjusting for sex, age, CCI, and vaccination history, using calendar time as the time axis. Hazard ratios with 95% CIs for SARS-CoV-2 infection were also estimated by age group (0–4, 5–17, 18–34, 35–64, 65–79, and ≥80 years), immunization status, and years since cancer diagnosis.

To assess the risk of severe disease or death, given the SARS-CoV-2 infection, a binary outcome was considered (severe or not severe) and the time since infection to hospitalization or death was not considered an outcome of interest. Multivariable logistic regression models were applied to estimate odds ratios (ORs) and 95% confidence intervals (CIs) for individuals with cancer compared with those without cancer, adjusting for age, sex, vaccination history, and CCI. Stratified analyses by age group (0–64, 65–79, and ≥80 years), immunization status, and years since cancer diagnosis are reported.

Infections occurring before 31 August 2020, were excluded from the analysis on disease severity since most cases of COVID-19 during the first wave were diagnosed within hospital settings in people with moderate-to-severe symptoms, resulting in a strong underestimation of mild and asymptomatic cases in the denominator.

All analyses were stratified by the pre-Omicron and Omicron pandemic periods.

STATA v. 18.0 was used for all analyses (StataCorp LLC, College Station, TX, USA).

## 3. Results

The study included 538,516 residents of the Reggio Emilia province, among whom 33,307 (6.2%) had a cancer diagnosis between 1996 and 2021. As shown in Table 1, cancer patients accounted for 9135 SARS-CoV-2 infections compared with 181,713 infections among individuals without cancer. The majority of infections occurred during the Omicron phase (January–September 2022). During the pre-Omicron phase (February 2020–December 2021) there were 2591 infections among cancer patients compared with 52,315 among non-cancer individuals. During the Omicron phase infections increased substantially in both groups (6544 vs. 129,398, respectively), consistent with greater viral transmissibility and new testing modalities like self-tests and antigen swabs.

Cancer patients were predominantly female (54%) and aged ≥ 65 years (64%), reflecting the demographic structure of the cancer survivor population. They also presented a higher Charlson comorbidity index (CCI), despite the exclusion of cancer from the score calculation in this study (Appendix A).

As reported in Table 2, the hazard ratio (HR) for SARS-CoV-2 infection was 1.00 (95% CI 0.96–1.05) during the pre-Omicron period and 1.08 (95% CI 1.05–1.11) during the Omicron period. Infection risk was slightly higher among cancer patients with a diagnosis in the previous two years (pre-Omicron HR 1.37, 95% CI 1.25–1.50; Omicron HR 1.35, 95% CI 1.28–1.44), while those with a diagnosis more than five years earlier had risks comparable to the general population.

As illustrated in Figure 1 and Appendix A, vaccination strongly reduced infection risk in both cancer and non-cancer groups. In the pre-Omicron phase, individuals who received three vaccine doses had a 92% risk reduction (HR 0.08, 95% CI 0.05–0.12), while prior infection conferred an 80% reduction (HR 0.20, 95% CI 0.10–0.31). The strongest protection was observed among individuals with hybrid immunity (prior infection + two vaccine doses), who exhibited a 97% risk reduction (HR 0.03, 95% CI 0.00–0.21).

Table 3 shows that cancer patients had greater odds of hospitalization or death compared with the general population once infected with SARS-CoV-2. The adjusted odds ratio (OR) for severe outcomes was 1.33 (95% CI 1.20–1.48) in the pre-Omicron period and 1.67 (95% CI 1.48–1.90) during Omicron. The excess risk was most pronounced in patients diagnosed < 2 years prior (OR 1.94 pre-Omicron; 3.38 Omicron), while those with a diagnosis more than 5 years earlier exhibited smaller excesses (OR 1.19 and 1.37, respectively).

In the pre-Omicron period, vaccination conferred strong protection against severe disease and death in both populations (Figure 2 and Appendix A). Among patients who received three or more doses, the odds of hospitalization or death were 60–80% lower than among unvaccinated individuals (ORs ranging from 0.2 to 0.4).

Results were substantially similar when a 14-day lag time from vaccine to protection was considered (Appendix A).

Overall, the absolute risk of severe COVID-19 declined sharply after December 2021, reflecting the lower virulence of circulating variants and extensive vaccine coverage.

Although the Omicron period was associated with partial immune escape, with reduced or null protection vs. infection, vaccine effectiveness against severe outcomes remained substantial in both the general population and in cancer patients. The combination of vaccination and prior infection conferred the highest level of protection, particularly during the Omicron period.

## 4. Discussion

### 4.1. Main Findings

In this population-based study from northern Italy, cancer patients showed a modest excess risk of infection and a persistently higher risk of severe outcomes compared to the general population. The increase in risk is particularly appreciable during the first two years after diagnosis. The excess of risk compared to the general population decreases with increasing age and with time since diagnosis.

The protective role of vaccination was strikingly consistent across cancer and non-cancer groups. While there was a strong protection of two doses in the pre-Omicron period for all, in the Omicron period the protection of the three doses schedule waned and only a mix of natural and vaccine immunization was protective vs. infection. On the other hand, vaccination with three doses gave protection vs. severe disease in the Omicron period.

Despite a similar protection, some differences between the cancer and general populations were appreciable. Vaccine efficacy was always lower in people with a recent cancer diagnosis, i.e., those more likely to be in the treatment phase. On the contrary, cancer patients who received a diagnosis more than 5 years earlier had an even larger protection, at least in the pre-Omicron period, than the general population, suggesting that in this group vaccination was probably associated with protective behaviours that further reduced the probability of infections.

### 4.2. Comparison with Previous Literature

Our findings are consistent with the multicenter cohorts investigated by Kuderer et al. and Yang et al. [1,3], who reported increased hospitalization and mortality among cancer patients, especially those receiving active treatment. Similar trends were observed in Johannesen et al. and Chavez-MacGregor et al. [5,8], emphasizing that the burden of comorbidities and immunosuppressive therapy are central drivers of poor prognosis. The present analysis extends these findings into the Omicron phase, showing that while absolute mortality decreased, the relative risk gap between cancer and non-cancer patients persisted or even increased.

In our population, the relative increase in risk of severe disease of cancer decreases with age. This is consistent with previous studies investigating the role of comorbidities increasing the risk of death in COVID-19 patients [15]. The reduction is not necessarily appreciable in terms of risk difference; in fact, the absolute risk sharply increases with age. Therefore, a smaller relative risk could correspond to a larger increase in risk difference.

The protective role of vaccination was strikingly consistent across cancer and non-cancer groups, corroborating evidence from Shroff et al. and Andrews et al. [7,11]. These results are consistent with other observations showing strong mRNA vaccine efficacy, even in immunocompromised hosts. Nevertheless, we observed a reduced effect of any kind of immunization, vaccine-induced, natural, or hybrid, in cancer patients with recent diagnosis. The reduced efficacy for infection could be confounded by the higher probability of testing in this group, which had frequent access to healthcare facilities. Furthermore, in the Omicron period the general population was mostly tested with antigenic tests that were less sensitive, particularly in the late period of infection, while cancer patients were still tested often with PCR tests, thus increasing the difference in the probability of detecting an infection. Other studies found reduced efficacy in some subpopulations of cancer patients undergoing specific therapies [16,17,18,19,20,21,22,23,24]. Recent evidence indicates that the excess COVID-19 risk in cancer patients is partly driven by vulnerability profiles rather than cancer alone. A recent systematic review (Hwang et al. [2], 2025) identified with high certainty male sex, older age, high comorbidity burden measured through a high Charlson comorbidity index (≥3), immunosuppression, chronic kidney disease, obesity, low performance status, and lung cancer as major prognostic factors for mortality. Mechanisms differ across outcomes: infection risk may increase through immune dysfunction and greater contact with healthcare services, while progression to severe disease is favoured by reduced physiological reserve, organ damage, and treatment-related toxicity.

The reduction in protective effect vs. severe disease cannot be explained by these biases in the probability and modality of testing. An explanation could be a misclassification of the cause of hospitalization or death in these patients, creating a background noise of outcomes attributed to COVID-19 but not caused by COVID-19. This necessarily reduces the measured efficacy, particularly in those patients who are frequently hospitalized due to other conditions, such as cancer patients. This phenomenon was particularly strong during the Omicron period, when COVID-19 was less severe and when the proportion of misclassification was necessarily higher. Nevertheless, in the Omicron period, we observed a small excess of severe COVID-19 for patients with recent diagnosis of cancer compared with the general population among the non-vaccinated, suggesting that this bias is present but not too large. The definition of severe COVID-19 relied on hospitalization with SARS-CoV-2 infection and COVID-19 death. Information about causes of hospitalization and discharge diagnoses was not available in the COVID-19 information systems. As a result, some hospitalisations may not have been directly caused by COVID-19, potentially introducing misclassification.

Despite the reassuring protection conferred by vaccination, patients under active therapy remain highly vulnerable, especially during waves driven by immune-evasive variants. Tailored preventive strategies, including priority access to booster doses, monoclonal prophylaxis, and reinforced infection control, remain essential for this subgroup. We were unable to stratify patients by cancer stage or by active anticancer treatment, as these variables are not consistently available in the cancer registry data for the entire cohort. Some of these conditions may be common risk factors of COVID-19 severity and cancer or facilitating conditions for COVID-19 infection and progression in cancer patients. Consequently, our categories based on time since diagnosis may still encompass heterogeneous clinical conditions, preventing us from identifying the intermediate factors playing a role in the COVID-19 and cancer interaction and eliminating some possible confounders.

Notably, the benefit of hybrid immunity observed here parallels results by Vicentini et al. [9] in the same population, highlighting that vaccination following natural infection provided the most durable protection against reinfection and severe disease in cancer patients.

## 5. Conclusions

In summary, this study showed that while COVID-19 vaccines substantially reduced infections and severe outcomes in both cancer and non-cancer populations, cancer patients, particularly those recently diagnosed, continued to face the highest risks. Hybrid immunity offered the greatest protection in both populations. During the Omicron phase, while risk of infection increased, the risk of severe disease declined, reflecting combined effects of vaccination, prior exposure, and variant attenuation, but the severity reduction was less pronounced for cancer patients.

Although vaccine efficacy was broadly similar between cancer and non-cancer groups, or even slightly less effective in cancer patients, the higher baseline risk of severe disease makes vaccination particularly important in this population. These findings underscore the importance of maintaining targeted vaccination policies, prioritizing boosters, and sustaining active surveillance for individuals with recent or ongoing cancer therapy.

## Figures and Tables

**Figure 1 vaccines-13-01223-f001:**
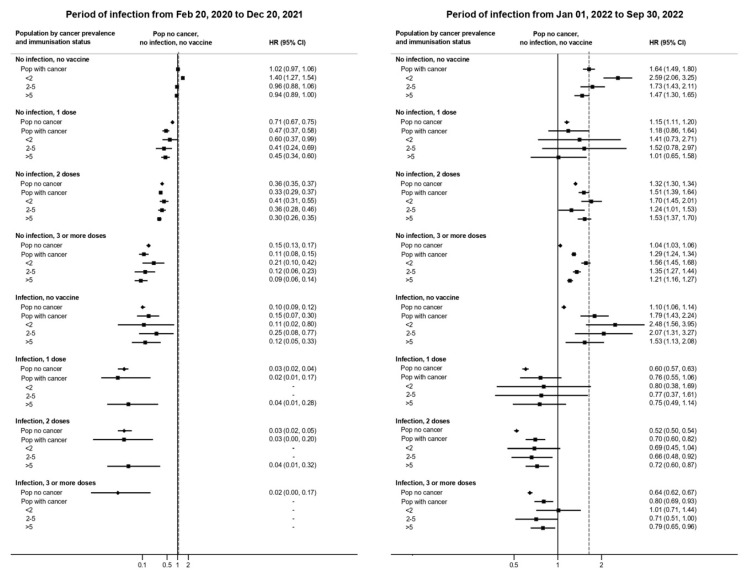
Risk of SARS-CoV-2 infection by cancer prevalence and immunization status adjusted for sex, age, and Charlson comorbidity index in the pre-Omicron and Omicron BA.1 periods, Reggio Emilia province, Italy, 20 February 2020–30 September 2022. Reinfections are included if they occurred at least 90 days after the previous infection; person-time in the 90 days after infection is excluded from the denominator. All covariates are time-dependent variables. The reference is the risk in the population without cancer, without previous infection, and not vaccinated (solid vertical line), and all hazard ratios (HRs) are relative to this value. The dashed vertical line corresponds to the HR of the group of people with cancer, not vaccinated, and no prior infection. Diamonds represent the risk of individuals without cancer, while squares represent the risk of individuals with cancer, with different vaccine and previous infection status. Three orders of comparison are possible in this figure: (**a**) compare diamonds to the solid line to evaluate how immunization status modifies infection risk separately in the non-cancer population; (**b**) compare squares to the dotted vertical line to evaluate how immunization status modifies infection risk separately in the cancer population (see Appendix A for HRs using the population with cancer, without previous infection, and not vaccinated as reference); (**c**) compare squares to diamonds to see the differences in people with cancer vs. people without cancer across immunization strata.

**Figure 2 vaccines-13-01223-f002:**
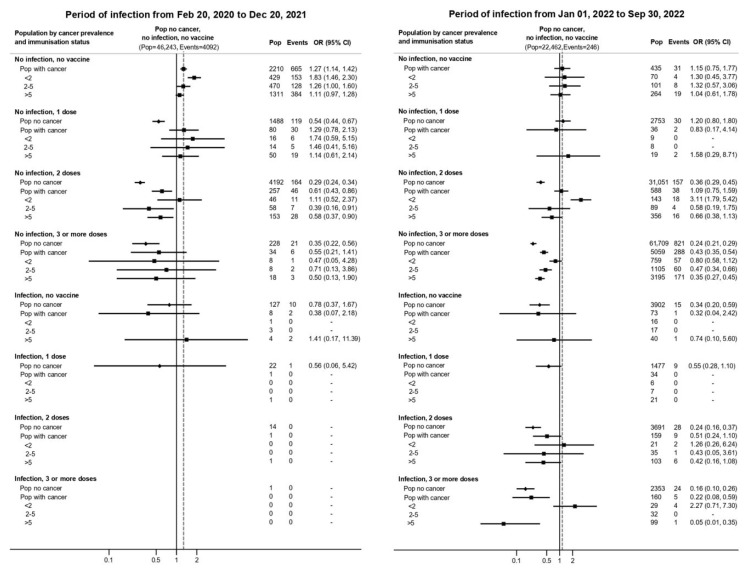
Risk of severe disease and death from COVID-19 by cancer prevalence and immunization status adjusted for sex, age, and Charlson comorbidity index in the pre-Omicron and Omicron BA.1 periods, Reggio Emilia province, Italy, 20 February 2020–30 September 2022. All covariates are considered at the time of infection. The outcome hospitalization or death could occur in the 45 days after infection or reinfections. The reference is the risk of people without cancer, without previous infection, and not vaccinated (solid vertical line), and all odds ratios (ORs) are relative to this value. The dashed vertical line corresponds to the OR of the group of people with cancer, not vaccinated, and no prior infection. Diamonds represent the risk of individuals without cancer, while squares represent the risk of individuals with cancer, with different vaccine and previous infection statuses. Three orders of comparison are possible in this figure: (**a**) compare diamonds to the solid line to evaluate how immunization status modifies infection risk separately in the non-cancer population; (**b**) compare squares to the dotted vertical line to evaluate how immunization status modifies infection risk separately in the cancer population (see Appendix A for ORs using population with cancer, without previous infection, and not vaccinated as reference); (**c**) compare squares to diamonds to see the different risks in people with cancer vs. people without cancer across immunization strata.

**Table 1 vaccines-13-01223-t001:** Cohort characteristics by history of cancer, Reggio Emilia province, Italy, 20 February 2020–30 September 2022.

	Population With Cancer					Population Without Cancer				
		Period of Infection		Period of Infection			Period of Infection		Period of Infection	
		From 20 February 2020 to 20 December 2021		From 1 January 2022 to 30 September 2022			From 20 February 2020 to 20 December 2021		From 1 January 2022 to 30 September 2022	
	*n* *	Person-Days	Infections	Person-Days	Infections	*n* *	Person-Days	Infections	Person-Days	Infections
Overall	26,928	18,665,758	2591	7,287,029	6544	511,588	333,498,175	52,315	123,371,182	129,398
Sex										
Male	12,067	8,326,317	1190	3,233,336	2812	253,329	165,256,721	25,865	61,568,939	60,125
Female	14,861	10,339,441	1401	4,053,693	3732	258,259	168,241,454	26,450	61,802,243	69,273
Age group										
0–4	10	3540	2	1100	0	21,241	5,323,859	1041	1,790,411	2447
5–17	113	73,584	14	27,223	41	70,367	47,820,086	8498	16,395,107	22,347
18–34	536	324,985	68	116,849	166	93,184	60,169,850	10,746	22,302,927	25,678
35–64	8958	5,855,449	956	2,234,436	2917	227,348	150,093,621	23,727	55,714,636	58,576
65–79	10,657	7,300,650	849	2,883,145	2289	67,820	46,891,575	5071	18,349,242	11,451
80+	6654	5,107,550	702	2,024,276	1083	31,628	23,199,184	3232	8,818,859	4229
Charlson comorbidity index										
0	21,777	15,539,301	2043	6,224,630	5609	491,821	321,732,201	50,165	119,394,475	125,929
1	2443	1,556,706	239	550,944	460	13,471	8,238,709	1382	2,873,016	2510
2	820	486,989	114	156,583	155	4151	2,383,848	486	767,292	682
3	1888	1,082,762	195	354,872	320	2145	1,143,417	282	336,399	277
Years from diagnosis										
<2		2,508,596	500	880,032	1053					
2–5		3,941,658	553	1,497,950	1394					
>5		12,215,504	1538	4,909,047	4097					
Immunization status										
No infection, no vaccine		11,841,178	2210	321,217	435		251,912,448	46,243	17,809,840	22,462
No infection, 1 dose		918,722	80	23,883	36		14,630,632	1488	1,094,708	2753
No infection, 2 doses		4,736,273	257	331,742	588		53,025,309	4192	14,500,761	31,051
No infection, 3 or more doses		567,032	34	5,571,101	5059		2,708,425	228	65,025,072	61,709
Infection, no vaccine		222,370	8	88,452	73		5,521,098	127	4,970,897	3902
Infection, 1 dose		228,024	1	32,247	34		3,739,067	22	1,765,645	1477
Infection, 2 doses		141,443	1	352,240	159		1,898,408	14	10,531,276	3691
Infection, 3 or more doses		10,716	0	566,147	160		62,788	1	7,672,983	2353

* Characteristics of the cohorts on 1 January 2020. Cancers diagnosed after this data are considered in the person-days.

**Table 2 vaccines-13-01223-t002:** Risk of SARS-CoV-2 infection by cancer prevalence adjusted for sex, age, Charlson comorbidity index, and immunization status in the pre-Omicron and Omicron BA.1 periods, Reggio Emilia province, Italy, 20 February 2020–30 September 2022.

	Risk of SARS-CoV-2 Infection					
	Period of Infection			Period of Infection		
	From 20 February 2020 to 20 December 2021			From 1 January 2022 to 30 September 2022		
	HR	95% CI		HR	95% CI	
**Cancer**						
No	1			1		
Yes	1.00	0.96	1.05	1.08	1.05	1.11
**Years from diagnosis**	-			-		
<2	1.37	1.25	1.50	1.35	1.28	1.44
2–5	0.98	0.90	1.07	1.12	1.06	1.18
>5	0.93	0.88	0.98	1.02	0.98	1.05
**0–4**						
**Cancer**						
No	1			1		
Yes	2.88	0.61	13.55	**-**	**-**	**-**
**5–17**						
**Cancer**						
No	1			1		
Yes	1.10	0.66	1.84	1.16	0.86	1.55
**18–34**						
**Cancer**						
No	1			1		
Yes	1.25	0.99	1.59	1.24	1.06	1.44
**35–64**						
**Cancer**						
No	1			1		
Yes	1.04	0.97	1.11	1.07	1.03	1.11
**65–79**						
**Cancer**						
No	1			1		
Yes	1.06	0.98	1.14	1.13	1.08	1.19
**80+**						
**Cancer**						
No	1			1		
Yes	0.96	0.96	1.05	1.07	1.01	1.13

Reinfections are included if they occurred at least 90 days after the previous infection; person-time in the 90 days after infection is excluded from the denominator. All covariates are time-dependent variables. HR, hazard ratio; CI, confidence interval.

**Table 3 vaccines-13-01223-t003:** Risk of severe disease and death from COVID-19 by cancer prevalence adjusted for sex, age, Charlson comorbidity index, and immunization status in the pre-Omicron and Omicron BA.1 periods, Reggio Emilia province, Italy, 20 February 2020–30 September 2022.

	Risk of Severe Disease and Death from COVID-19									
	Period of Infection					Period of Infection				
	From 20 February 2020 to 20 December 2021					From 1 January 2022 to 30 September 2022				
	*n*	Events	OR	95% CI		*n*	Events	OR	95% CI	
**Cancer**										
No	52,315	4407	1			129,398	1330	1		
Yes	2591	749	1.33	1.20	1.48	6544	374	1.67	1.48	1.90
**Years from diagnosis**			-					-		
<2	500	171	1.94	1.57	2.41	1053	85	3.38	2.66	4.31
2–5	553	142	1.28	1.02	1.59	1394	73	1.68	1.30	2.17
>5	1538	436	1.19	1.05	1.36	4097	216	1.37	1.17	1.60
**0–64 ^#^**										
**Cancer**										
No	44,012	1711	1			110,855	330	1		
Yes	1040	107	1.42	1.14	1.77	2812	44	3.16	2.22	4.50
**65–79**										
**Cancer**										
No	5071	1228	1			12,827	327	1		
Yes	849	284	1.36	1.16	1.61	2279	108	1.55	1.23	1.96
**80+**										
**Cancer**										
No	3232	1468	1			5716	673	1		
Yes	702	358	1.13	0.95	1.33	1453	222	1.48	1.24	1.75

All covariates are considered at the time of infection. The outcome hospitalization or death could occur in the 45 days after infection. Reinfections are included if they occurred at least 90 days after the previous infection. ^#^ The age categories were merged due to the low number of events in the 0–4, 5–17, and 18–34 age groups: only one event in the cancer population in both periods. OR, odds ratio; CI, confidence interval

## Data Availability

Data can be requested from the authors after an approval of a protocol of analysis by the Area Vasta Emilia Nord Ethics Committee.

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
