# Peer review of "Assessing COVID-19 Infection and Severe Disease Risk in Cancer Patients and Survivors: The Role of Vaccination Status, Circulating Variants, and Comorbidities—A Population-Based Study in Northern Italy"

_vaccines, 2025, doi:10.3390/vaccines13121223_

Round 1
Reviewer 1 Report
Comments and Suggestions for Authors
This manuscript presents COVID-19 cases among cancer patients using population-based data from the Reggio Emilia province in Italy. The study links multiple registries, including the Reggio Emilia Cancer Registry, COVID-19 surveillance, vaccination, and hospitalisation databases, and spans from the ancestral SARS-CoV-2 strain to the Omicron variants. The inclusion of data from the early pandemic period, before vaccine rollout, makes this work particularly valuable.
The findings provide comprehensive and insightful evidence on the effects of vaccination and hybrid immunity among cancer patients and survivors. The manuscript is well-written, and the findings are of significant public health and clinical importance.
Comments.
1. Clarification of cancer cases:
The Methods section in line 114 states that the Reggio Emilia Cancer Registry provided information on "all incident malignant tumors". To allow for a more precise interpretation of the results, the authors should explicitly clarify the scope of this definition. Does this include both solid tumours and haematological malignancies?
This distinction is critical, as haematological malignancies often confer a different (and typically higher) risk of immune vulnerability, impaired vaccine response, and severe COVID-19 outcomes compared to most solid tumours.
2. Definition of severe COVID-19:
Was the definition of severe COVID-19 based solely on hospitalisation, regardless of pneumonia status, ICU admission, or other clinical criteria?
Some hospitalisations might have occurred for non-COVID-related causes coinciding with SARS-CoV-2 positivity.
If detailed clinical data are unavailable, consider acknowledging this limitation explicitly.
3. Follow-up procedure:
The description in lines 160–162 appears ambiguous: “Follow-up began on 20 February 2020 and ended at the first occurrence of infection, death, or 30 September 2022…” and later “For reinfections, follow-up started 90 days after the first positive test…”
Normally, follow-up should end only upon death, loss to follow-up, or withdrawal.
All participants alive (including COVID-19 survivors) should remain under observation until the study’s end date (30 September 2022). Please clarify or rephrase accordingly.
4. Age grouping and paediatric consideration:
The 0–34-year age group combines paediatric and young-adult populations, which differ biologically and in vaccine rollout timing.
Children and adolescents (<18) were COVID-19 vaccinated later due to delayed regulatory approval.
Paediatric cancers are predominantly haematological, CNS, bone or spinal, unlike adult solid tumours.
This grouping could confound comparisons of infection or severity risk. Consider discussing this limitation or providing a sensitivity analysis.
5. Cancer stage and treatment status:
The stratification by "time since diagnosis" (<2, 2-5, >5 years) is a useful and well-justified proxy for disease/treatment status. However, this group is still heterogeneous. The analysis would be substantially strengthened if data on active treatment status (e.g., active chemotherapy, immunotherapy, or other immunosuppressive therapies) or cancer stage were available. Patients with advanced-stage disease or those on active systemic therapy are at markedly higher risk. If these data are unavailable, this should be noted as a key limitation in the Discussion.
6. Charlson Comorbidity Index (CCI) calculation:
The Results section notes that cancer was excluded from the CCI computation to avoid circularity; this is appropriate and commendable.
However, this methodological decision should appear in the Materials and Methods section (under “Covariates” or “Statistical analysis”) rather than in Results, ensuring transparency from the outset.
7. Vaccination timing and infection classification:
Please clarify whether infection dates were compared with vaccination dates to ensure that immunity had time to develop.
Immune protection typically peaks 7–14 days post-vaccination. Infections occurring within this interval (<7 days) should not be considered “vaccinated” events, as vaccine protection would not yet be effective. Misclassification here could bias vaccine-effect estimates.
Typos.
1. Line 219: “diignosis” → “diagnosis.”
Author Response
Reviewer 1
This manuscript presents COVID-19 cases among cancer patients using population-based data from the Reggio Emilia province in Italy. The study links multiple registries, including the Reggio Emilia Cancer Registry, COVID-19 surveillance, vaccination, and hospitalisation databases, and spans from the ancestral SARS-CoV-2 strain to the Omicron variants. The inclusion of data from the early pandemic period, before vaccine rollout, makes this work particularly valuable.
The findings provide comprehensive and insightful evidence on the effects of vaccination and hybrid immunity among cancer patients and survivors. The manuscript is well-written, and the findings are of significant public health and clinical importance.
Comments.
- Clarification of cancer cases:
The Methods section in line 114 states that the Reggio Emilia Cancer Registry provided information on "all incident malignant tumors". To allow for a more precise interpretation of the results, the authors should explicitly clarify the scope of this definition. Does this include both solid tumours and haematological malignancies?
This distinction is critical, as haematological malignancies often confer a different (and typically higher) risk of immune vulnerability, impaired vaccine response, and severe COVID-19 outcomes compared to most solid tumours.
RE: Thank you for this important comment.
In the new version, we clarified that the Reggio Emilia Cancer Registry follows international IARC/IACR standards and therefore records all incident malignant neoplasms, including both solid tumours and haematological malignancies, as defined by ICD-O-3 behaviour code /3.
In line with international guidelines, the Registry also includes in incidence the non-malignant (benign and uncertain behaviour) tumours of the brain and other central nervous system, and the non-malignant tumours of the urinary bladder (ICD-O-3 behaviour codes /0 and /1 for these specific sites), due to their clinical relevance and the need for international comparability.
For this study, the cancer cohort included all incident malignant tumours, excluding non-melanoma skin cancers. In the presence of multiple primaries, the latest tumour was selected for metachronous cases, while for synchronous tumours the most severe tumour was retained, according to standard cancer registration rules. We added a paragraph to the line 119.
- Definition of severe COVID-19:
Was the definition of severe COVID-19 based solely on hospitalisation, regardless of pneumonia status, ICU admission, or other clinical criteria?
Some hospitalizations might have occurred for non-COVID-related causes coinciding with SARS-CoV-2 positivity.
If detailed clinical data are unavailable, consider acknowledging this limitation explicitly.
RE: We thank the reviewer for this observation. The definition of severe COVID-19 in our study was indeed based on hospitalisation with SARS-CoV-2 infection, as more detailed clinical information (e.g., pneumonia status, oxygen requirement, Intensive Care Unit admission) is not systematically captured in our COVID-19 information systems data sources. We agree that some hospitalizations may not have been directly caused by COVID-19, and this potential misclassification was already acknowledged as a study limitation. Unfortunately, Intensive Care Unit data were not reliable enough to refine severity classification, as during specific phases of the pandemic several emergency ICU units were activated and not consistently traceable within the routine information systems. For these reasons, hospitalization was the only consistently available and valid indicator of disease severity across the entire study population and time period. To address the reviewer’s point, we have reinforced this explanation in the Methods (line 150) and Discussion section (line 390).
- Follow-up procedure:
The description in lines 160–162 appears ambiguous: “Follow-up began on 20 February 2020 and ended at the first occurrence of infection, death, or 30 September 2022…” and later “For reinfections, follow-up started 90 days after the first positive test…”
All participants alive (including COVID-19 survivors) should remain under observation until the study’s end date (30 September 2022). Please clarify or rephrase accordingly.
RE: We rephrased and added that the follow up ended at the end of study for those who did not die. We also better specified that we excluded a 90-day period after infection from the at risk time for re-infection.
We better explained how we classified the hospitalizations and deaths related to an infection.
- Age grouping and paediatric consideration:
The 0–34-year age group combines paediatric and young-adult populations, which differ biologically and in vaccine rollout timing.
Children and adolescents (<18) were COVID-19 vaccinated later due to delayed regulatory approval.
Paediatric cancers are predominantly haematological, CNS, bone or spinal, unlike adult solid tumours.
This grouping could confound comparisons of infection or severity risk. Consider discussing this limitation or providing a sensitivity analysis.
RE we thank the reviewer for this suggestion. In the new version, we split the 0-34 group into three separate classes: the children below 5, who were eligible for vaccination only since December 2022; 5 to 17 as pediatric and adolescent population and young adults 18-34.
All the tables have been reframed accordingly, but that for severe disease ORs since there are too few events in the younger ages. Nevetheless the row numbers for the detailed age grouls are reported in supplementary tables.
- Cancer stage and treatment status:
The stratification by "time since diagnosis" (<2, 2-5, >5 years) is a useful and well-justified proxy for disease/treatment status. However, this group is still heterogeneous. The analysis would be substantially strengthened if data on active treatment status (e.g., active chemotherapy, immunotherapy, or other immunosuppressive therapies) or cancer stage were available. Patients with advanced-stage disease or those on active systemic therapy are at markedly higher risk. If these data are unavailable, this should be noted as a key limitation in the Discussion.
Re: We agree with the reviewer that heterogeneity remains within the “time since diagnosis” categories, and that information on active oncological treatment and cancer stage would allow a more refined stratification of risk. Unfortunately, these data are not consistently available in our COVID-19 information systems and cancer registry sources for the entire population and period considered. Treatment data are not available in cancer registry, and stage at diagnosis is only available for a few tumour types and periods. It is worth noting that active anticancer treatments may be conceptually intermediate factors on the causal pathway between cancer and severe COVID-19 outcomes, rather than confounders. Treatments are largely determined by cancer type, severity, and time since diagnosis. Adjusting for an intermediate variable could introduce overadjustment and underestimate the total effect of cancer on COVID-19 outcomes. We agree that patients with advanced disease or undergoing active systemic treatment may have a different risk profile and it would be essential to stratify analyses accordingly to understand why and which cancer patients have higher risk of infection and severe covid-19. We have highlighted this limitation in the discussion (lines 390-399).
- Charlson Comorbidity Index (CCI) calculation:
The Results section notes that cancer was excluded from the CCI computation to avoid circularity; this is appropriate and commendable.
However, this methodological decision should appear in the Materials and Methods section (under “Covariates” or “Statistical analysis”) rather than in Results, ensuring transparency from the outset.
Re: We thank the reviewer for this helpful observation. As noted, cancer was excluded from the computation of the Charlson Comorbidity Index to avoid circularity when analysing outcomes by cancer status. Although the CCI definition is already described in the Results section, we agree that the explicit clarification regarding the exclusion of cancer should be stated. We revised the Materials and Methods section to clearly specify that cancer was intentionally omitted from the CCI score for this study (line 145)
- Vaccination timing and infection classification:
Please clarify whether infection dates were compared with vaccination dates to ensure that immunity had time to develop.
Immune protection typically peaks 7–14 days post-vaccination. Infections occurring within this interval (<7 days) should not be considered “vaccinated” events, as vaccine protection would not yet be effective. Misclassification here could bias vaccine-effect estimates.
Re: We thank the reviewer. We added a sensitivity analysis adopting the definition of protected and partially protected based on the timing of first an second dose used by Fabiani et al BMJ 2022, i.e. until 14 days after first dose the subject is not protected (as those without any dose), then is partially protected, in our study reported as only one dose, until 14 days after the second dose, when the subject is considered completely vaccinated, in our study 2 doses. Third and further doses were considered as booster and no lag time was adopted.
Typos.
1. Line 219: “diignosis” → “diagnosis.”
RE: we thank the reviewer.
Reviewer 2 Report
Comments and Suggestions for Authors
Thank you kindly for the opportunity to review your manuscript Assessing COVID vaccination in patients with cancer. I am impressed by your work. I have a few comments for you to consider - use those that you find useful in improving your manuscript and ignore those that you don't believe are helpful.
Please see line 59: "emergence of circulating SARS-CoV-2 variants" Did you intend to indicate that these were NEW variants or merely that there was more than one?
See line 104: "whether they had..." Generally when this wording is chosen, the end of the statement is "or not". It may be easier here to say "Whether or not" they had ...
See line 166: There is something missing in this sentence. "hospitalizations were included if occurred up to 28 October..." If they occurred?
See line 168: I believe there is a typo in here somewhere. As written pre-Omicron ends on 20 December and Omicron begins on 1 January - - what happens to the days 21 December to 31 December? They are not in either group.
See line 200: Is this first sentence author instructions that can be removed from your manuscript?
See line 214: There is something missing in this statement "higher CCI, despite cancer was not included in the computation..."
Lines 220 and 223 have some spacing errors that will likely be corrected by the journal editor
Author Response
Reviewer 2
Thank you kindly for the opportunity to review your manuscript Assessing COVID vaccination in patients with cancer. I am impressed by your work. I have a few comments for you to consider - use those that you find useful in improving your manuscript and ignore those that you don't believe are helpful.
Please see line 59: "emergence of circulating SARS-CoV-2 variants" Did you intend to indicate that these were NEW variants or merely that there was more than one?
Re: thank you for this comment. We have clarified the text to specify that we refer to the emergence and rapid replacement of successive SARS-CoV-2 variants, rather than simply the coexistence of multiple variants. In particular, the Omicron lineage rapidly became dominant, almost completely replacing the Delta variant within a few weeks. To support this, we now cite national genomic surveillance data showing that by 3 January 2022, Omicron already accounted for 76.9–80.2% of sequenced infections in Italy, and Delta had fallen below 6% by 17 January 2022 (Stefanelli et al., Euro Surveillance 2022). To improve the reader’s understanding, we have also added a supplementary table reporting the temporal distribution of variants in our region based on local surveillance data, showing the progressive transition from Delta to Omicron
We have revised the sentence in the Introduction accordingly
See line 104: "whether they had..." Generally when this wording is chosen, the end of the statement is "or not". It may be easier here to say "Whether or not" they had ...
RE: Thank you for the suggestion. We agree that the sentence becomes clearer when formulated using “whether or not”, and we have revised the text accordingly.
10 See line 166: There is something missing in this sentence. "hospitalizations were included if occurred up to 28 October..." If they occurred?
RE: Thank you for pointing this out. We agree that the sentence was grammatically incomplete. The text has been revised
See line 168: I believe there is a typo in here somewhere. As written pre-Omicron ends on 20 December and Omicron begins on 1 January - - what happens to the days 21 December to 31 December? They are not in either group.
Re: we thank the reviewer for this careful observation. The gap between 21 and 31 December 2021 was intentionally excluded from the analysis because it represented a transitional period in which Omicron was rapidly emerging but had not yet fully replaced previous variants. To avoid misclassification of infections by predominant variant period, we defined the pre-Omicron period as ending on 20 December 2021 and the Omicron period as starting on 1 January 2022, explicitly excluding the transitional days in between, as stated in the Methods section (Follow-up). We changed the method section to increase clarity (line 190-191)
See line 200: Is this first sentence author instructions that can be removed from your manuscript?
RE: Ok thanks
See line 214: There is something missing in this statement "higher CCI, despite cancer was not included in the computation..."
RE: Thanks, we rephrased
Lines 220 and 223 have some spacing errors that will likely be corrected by the journal editor
RE: we thank the reviewer, we tried to edit all the typos
Reviewer 3 Report
Comments and Suggestions for Authors
This is a population-based cohort in linkage with the local cancer registry to the COVID-19 infection and disease severity among cancer patients and survivals. The study is overall well designed, however, there are a few issues the authors should consider for improvement.
- The observed association of higher infection and severity of diseases might be attributable to shared risk factors of both covid-19 and cancer. For example, prior research reported risk factors associated with covid-19 infection fatality rates and elderly people are more susceptible to high infection fatality rates. They authors should fully discuss all shared risk factors and the possibility that the observed higher infection and severity of diseases are due to those factors.
- Reference 1 is missing. The authors did not delete the editorial languages at the beginning of some paragraphs. Please check and add back.
- Table 2 use HR to model the risk of covid-19 infection by cancer status and age groups. However, Table 3 uses OR to model the severity and death from covid-19. Can you please elaborate the statistical method you select for the analysis in the Method?
- Figure 3 and 4 are crowded and a bit confusing. From my understanding, you may want to compare the risk of covid-19 infection/mortality among cancer patients and general population in each immunization status, which means that the non-cancer groups should be the reference group by each status. However, you use only "no infection, vaccine and no cancer" as the reference group for the whole group. This makes it harder to differentiate the effect of immunization and cancer on covid-19 infection and mortality. I suggest to separately test the effect of cancer/no-cancer, and vaccine/no-vaccine. In this way, you can make the Figure clearer and easier to compare within groups.
- The Discussion is not adequate. As I mentioned before, the authors need to read the previous literature and fully discuss the risk factors of infection fatality rates of covid-19 and cancer, which can help you have a clear idea about the causal relationship. You should consider adjustment for these common risk factors. For the vaccine effect, you could also consider add more analysis to understand the effect of vaccination by cancer status.
Author Response
Reviewer 3
This is a population-based cohort in linkage with the local cancer registry to the COVID-19 infection and disease severity among cancer patients and survivals. The study is overall well designed, however, there are a few issues the authors should consider for improvement.
- The observed association of higher infection and severity of diseases might be attributable to shared risk factors of both covid-19 and cancer. For example, prior research reported risk factors associated with covid-19 infection fatality rates and elderly people are more susceptible to high infection fatality rates. They authors should fully discuss all shared risk factors and the possibility that the observed higher infection and severity of diseases are due to those factors.
Re: we thank the reviewer for this relevant and constructive observation. We fully agree that cancer patients and survivors share several vulnerability pathways which may partially explain both the higher probability of SARS-CoV-2 infection and the excess severity once infected. In the initial version of the manuscript we mainly focused on overarching mechanisms (e.g., treatment-related immunosuppression, transplant-related immune dysfunction, or direct tumour-mediated organ impairment). Following the reviewer’s suggestion, we have now expanded this section by integrating more granular evidence from recent literature, including a systematic review using GRADE approach (Hwang et al., Int J Cancer 2025).
In particular, we distinguish factors that increase risk of infection, such as disease or treatment-related immunosuppression, frequent healthcare contacts and diagnostic procedures even during restrictive pandemic phases, while recognizing the potential mitigating mechanisms, including earlier access to vaccination and greater adherence to protective behaviours (e.g.isolation, risk-avoidance).
Conversely, determinants of severe disease once infected include the depth of immunosuppression, reduced physiological reserve, direct organ damage from cancer, treatment-induced toxicity (cardiac, renal and hematological impairment), low performance status. The systematic review we now cite reports high-certainty associations between mortality and several of these factors, including male sex, older age, comorbidity burden (CCI≥3), immunosuppression, chronic kidney disease, obesity, low ECOG performance status, lung cancer and severe clinical presentation.
Our models already adjust for major shared determinants such as age, sex, comorbidities and vaccination status/timing. However, we acknowledge that additional prognostic domains, cancer stage, active systemic therapy, degree of frailty, transplant status or immune reserve, are not available in our COVID-19 information systems. A dedicated paragraph has been added in the Discussion to reflect these points more explicitly (line 390). Moreover, we have added one sentence in the Introduction to anticipate physiological reserve, frailty status and organ damage with a reference (line 48).
Reference 1 is missing. The authors did not delete the editorial languages at the beginning of some paragraphs. Please check and add back.
Re: ok
- Table 2 use HR to model the risk of covid-19 infection by cancer status and age groups. However, Table 3 uses OR to model the severity and death from covid-19. Can you please elaborate the statistical method you select for the analysis in the Method?
RE: we thank the reviewer for this comment, giving us the opportunity to better explain how we modelled the probability of infection and the probability of having a severe disease given that one has been infectect.
The different denominators and time horizons of the two analyses led us to choose different models. For the probability of infection, we preferred a Cox proportional hazard survival analysis, with time-dependent variables and accounting for the different baseline risk in different calendar time periods. For the probability of having a severe disease given the infection, we used a logistic model, since the time elapsed from infection to hospitalisation or death is not an outcome of interest, and all variables were considered at the time of infection.
We added a sentence to explain the different denominator and outcome in the method section
Figure 3 and 4 are crowded and a bit confusing. From my understanding, you may want to compare the risk of covid-19 infection/mortality among cancer patients and general population in each immunization status, which means that the non-cancer groups should be the reference group by each status. However, you use only "no infection, vaccine and no cancer" as the reference group for the whole group. This makes it harder to differentiate the effect of immunization and cancer on covid-19 infection and mortality. I suggest to separately test the effect of cancer/no-cancer, and vaccine/no-vaccine. In this way, you can make the Figure clearer and easier to compare within groups.
RE: we thank the reviewer for this very thoughtful comment. We agree that Figures 3 and 4 are visually dense, and that the current choice of reference category may make it less intuitive to disentangle the effects of immunization status and cancer status. Our intention, however, was to allow readers to view all comparisons simultaneously, using a single, consistent reference group. This approach facilitates an overall interpretation of how cancer and immunization jointly modify the risk of SARS-CoV-2 infection and COVID-19 mortality. We also acknowledge that this structure may reduce immediate clarity for the reader. In line with the reviewer’s helpful suggestion, we have revised the figure caption to guide the reader across all the possible comparisons. We hope this improved the readability while maintaining the analytical structure of the figures.
The figure legend are now:
"Figure 1. Risk of SARS-CoV-2 infection according to cancer diagnosis and immunisation status, adjusted for age, sex, and Charlson Comorbidity Index, in the pre-Omicron and Omicron BA.1 periods, Reggio Emilia Province, Italy (20 Feb 2020–30 Sep 2022).
Re-infections were included if they occurred ≥90 days after the previous infection; person-time within 90 days of infection was excluded. All covariates are treated as time-varying.
The reference is the risk of people without cancer, without previous infection and not vaccinated (solid vertical line), and all HR are relative to this value.
The dashed vertical line corresponds to the HR of people with cancer, not vaccinated, no prior infection group.
Diamonds represent the risk of individuals without cancer, while squares represent the risk of individuals with cancer, with different vaccine and previous infection status.
Three orders of comparison are possible in this figure:
- Compare diamonds to the solid line to evaluate how immunisation status modifies infection risk separately in non-cancer population.
- Compare squares to the dotted vertical line to evaluate how immunisation status modifies infection risk separately in cancer population.
- Compare squares to diamonds to see the different in people with cancer vs. people without cancer across immunization strata."
"Figure 2. Risk of severe COVID-19 or death by cancer status and immunisation, adjusted for age, sex and Charlson Comorbidity Index, in the pre-Omicron and Omicron BA.1 periods, Reggio Emilia Province, Italy (20 Feb 2020–30 Sep 2022).
Outcomes could occur within 45 days from infection. Re-infections were included if they occurred ≥90 days after the previous one. All covariates are considered at the time of infection.
The reference is the risk of people without cancer, without previous infection and not vaccinated (solid vertical line), and all OR are relative to this value.
The dashed vertical line corresponds to the OR of people with cancer, not vaccinated, no prior infection group.
Diamonds represent the risk of individuals without cancer, while squares represent the risk of individuals with cancer, with different vaccine and previous infection status.
Three orders of comparison are possible in this figure:
- Compare diamonds to the solid line to evaluate how immunisation status modifies infection risk separately in non-cancer population.
- Compare squares to the dotted vertical line to evaluate how immunisation status modifies infection risk separately in cancer population.
- Compare squares to diamonds to see the different risk in people with cancer vs. people without cancer across immunization strata."
We also added two tables in the supplementary material (supplemental table 7 and 8) only for the population with cancer where the reference is the population without COVID-19 infection and without vaccine
The Discussion is not adequate. As I mentioned before, the authors need to read the previous literature and fully discuss the risk factors of infection fatality rates of covid-19 and cancer, which can help you have a clear idea about the causal relationship. You should consider adjustment for these common risk factors. For the vaccine effect, you could also consider add more analysis to understand the effect of vaccination by cancer status.
RE: We thank the reviewer for this important comment. We agree that several shared risk factors (e.g., age, comorbidities, frailty) and intermediate factors (such as cancer stage, recent treatments, or immune impairment) contribute to the higher infection fatality risk observed among cancer patients. We have strengthened the Discussion by explicitly acknowledging the complexity of these causal pathways and by summarising the relevant evidence from previous literature on the determinants of severe COVID-19 outcomes in people with cancer.
Regarding adjustment for these common risk factors, our analyses already account for age, sex, and comorbidities through the Charlson Comorbidity Index, and for vaccination status and timing, which cover some of the major determinants of COVID-19 severity. However, we recognise that other important variables, such as cancer stage, ongoing systemic treatments, frailty indicators, or behavioural factors, are not fully available in our COVID-19 information systems sources and therefore cannot be included in the adjustment set. As these factors lie partly on the causal pathway between cancer and COVID-19 outcomes, controlling for them could also introduce over-adjustment; nevertheless, we explicitly acknowledge their absence as a limitation. We added a paragraph in the discussion.
Round 2
Reviewer 1 Report
Comments and Suggestions for Authors
Thank you for thoroughly addressing the concerns I raised in your previous submission. The changes made have improved the clarity and overall quality of the manuscript.
Reviewer 3 Report
Comments and Suggestions for Authors
Thanks for addressing my previous comments.